# Visual Elements Mining as Prompts for Instruction Learning for Target-Oriented Multimodal Sentiment Classification

**Bin Yang** and **Jinlong Li**
University of Science and Technology of China
jcbx@mail.ustc.edu.cn, jlli@ustc.edu.cn

## Abstract

Target-oriented Multimodal Sentiment Classification (TMSC) aims to incorporate visual modality with text modality to identify the sentiment polarity towards a specific target within a sentence. To address this task, we propose a Visual Elements Mining as Prompts (VEMP) method, which describes the semantic information of visual elements with Text Symbols Embedded in the Image (TSEI), Target-aware Adjective-Noun Pairs (TANPs) and image scene caption, and then transform them into prompts for instruction learning of the model Tk-Instruct. In our VEMP, the text symbols embedded in the image may contain the textual descriptions of fine-grained visual elements, and are extracted as input TSEI; we extract adjective-noun pairs from the image and align them with the target to obtain TANPs, in which the adjectives provide emotional embellishments for the relevant target; finally, to effectively fuse these visual elements with text modality for sentiment prediction, we integrate them to construct instruction prompts for instruction-tuning Tk-Instruct which possesses powerful learning capabilities under instructions. Extensive experimental results show that our method achieves state-of-the-art performance on two benchmark datasets. And further analysis demonstrates the effectiveness of each component of our method[1].

## 1 Introduction

Target-oriented Multimodal Sentiment Classification (TMSC) is a fine-grained sentiment analysis task towards multimodal data, which aims to classify the sentiment polarity towards the given opinion target (a word or a phrase) in a sentence-image pair. For example, we should infer that the sentiment polarity towards the target term "Cleveland Cavaliers" is negative with the provided sentence and image in Figure 1(a).

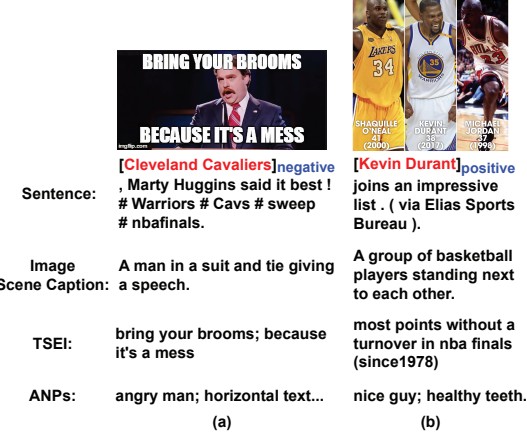

| | | |
|---|---|---|
| **Sentence:** | [Cleveland Cavaliers]negative, Marty Huggins said it best ! # Warriors # Cavs # sweep # nbafinals. | [Kevin Durant]positive joins an impressive list . ( via Elias Sports Bureau ) |
| **Image Scene Caption:** | A man in a suit and tie giving a speech. | A group of basketball players standing next to each other. |
| **TSEI:** | bring your brooms; because it's a mess | most points without a turnover in nba finals (since1978) |
| **ANPs:** | angry man; horizontal text... | nice guy; healthy teeth... |
| | (a) | (b) |

Figure 1: Two examples of the TMSC task. TSEI: text symbols embedded in the image. ANPs: adjective-noun pairs extracted from the image.

With the boom of multimodal multimedia (e.g., tweets) on the Internet, TMSC has been a valuable issue and has aroused a wave of research enthusiasm. Early work (Yu and Jiang, 2019; Wang et al., 2021; Zhang et al., 2021) focused on utilizing attention mechanism to fuse text features and visual features. Recently, Khan and Fu (2021) translated the image into textual image scene caption, which aligned different modalities with heightened interpretability since the image is represented in natural language. We concentrate on this new research formula to project visual modality to text modality for cross-modal fusion.

However, how to adequately mine visual elements to get a comprehensive utilization of visual semantic information remains an unsolved challenge. For instance, image scene caption has two deficiencies: (1) Generally, the image scene caption lacks detailed depiction of the image, which means information loss during the image-to-text translation process. As shown in Figure 1, the image scene captions are just coarse overviews of scenes in images. (2) It mostly involves neutral vo-

---

[1]Our code is publicly available at https://github.com/long8181/VEMP.

cabulary and lacks explicit affective guidance for sentiment prediction. As depicted in Figure 1, the image scene captions are objective narratives without emotional characterization. To handle these problems, Yang et al. (2022) tried to introduce facial emotion yet their method is constrained to images with facial expressions and thereby lacks generalizability. Therefore, we strive for more universal and efficient solution and propose a Visual Elements Mining as Prompts (VEMP) method.

Firstly, we extract Text Symbols Embedded in the Image (TSEI) to complement image scene caption with fine-grained details. We notice that in the datasets Twitter-2015 and Twitter-2017 (Zhang et al., 2018; Lu et al., 2018) more than 50% of samples' corresponding images contain TSEI, which can contribute to TMSC. As delineated in Figure 1, the TSEI "because it's a mess" are pivotal to judge the "negative" sentiment polarity towards the target "Cleveland Cavaliers" while the TSEI "most points without a turnover in nba finals (since1978)" are crucial to infer the "positive" sentiment polarity towards to the target "Kevin Durant". Therefore, in our VEMP, we attach importance to TSEI and employ Optical Character Recognition (OCR) technique to extract them. In order to denoise the extracted results for improved usability, we have executed elaborate post-processing refinements, such as spelling correction, grammar filtering, etc.

Secondly, we design a Target-aware Adjective-Noun Pairs (TANPs) extraction subtask to capture fine-grained and emotional visual content. Existing research has demonstrated the efficacy of ANPs in TMSC (Zhao et al., 2022), which directly employed ANPs to assist attention-based cross-modal features fusion. From another perspective, as depicted in Figure 1, we observe that the nouns are detailed objects in images and the adjectives imply emotional tendencies, which fill the above two gaps of image scene caption. Thus, we extract ANPs from the image as supplements to image scene caption. Besides, we align ANPs with the target to obtain TANPs since ANPs relevant to the target are helpful while the others introduce noises.

Finally, we utilize TSEI, TANPs, image scene caption and the sentence in the sentence-image pair to construct instruction prompts for instruction-tuning the model Tk-Instruct (Wang et al., 2022b), inspired by previous success of in-

struction learning across a variety of tasks (Zhang and Chai, 2021; Mishra et al., 2022; Ouyang et al., 2022). Our instruction prompt consists of a task definition, three examples with "positive", "negative" and "neutral" sentiment labels respectively, and a to-be-classified sample.

Our contributions are as follows:

(1) We propose a Visual Elements Mining as Prompts (VEMP) model, which mines TSEI, TANPs and image scene caption as textual descriptions of the image, leading to a more comprehensive understanding of visual modality. To the best of our knowledge, we are the first to take TSEI into consideration for TMSC. Moreover, VEMP is applicable to all types of images and exhibits universality.

(2) Our VEMP adopt the instruction learning paradigm and achieves state-of-the-art performance on benchmark datasets Twitter-2015 and Twitter-2017, which demonstrates the superiority of our method.

(3) We conduct experiments on GPT-4 with zero-shot and few-shot settings to explore its potential for TMSC.

## 2 Related Work

### 2.1 Target-oriented Multimodal Sentiment Classification

Target-oriented Multimodal Sentiment Classification (TMSC), an extension task of Aspect-Based Sentiment Analysis (ABSA) tailored for multimodal data, has attracted extensive attention lately due to the increase of multimodal information on the Web. Xu et al. (2019) and Yu et al. (2019) applied attention mechanism based on LSTM for cross-modal fusion. Yu and Jiang (2019), Wang et al. (2021) and Zhang et al. (2021) further chose BERT-based architecture to model the cross-modal interactions. Khan and Fu (2021) creatively proposed a novel method to translate the image into textual image scene caption, which effectively aligns multimodal features from different feature spaces. However, image scene caption lacks detailed and emotional depiction of the image. To address that, Yang et al. (2022) captured facial emotion from the image, which is limited to images containing facial expressions. Zhao et al. (2022) proposed a knowledge-enhanced framework (KEF) which employed adjective-noun pairs (ANPs) from the image to assist attention-based models. Ling et al. (2022) proposed a

Vision-Language Pre-training framework (VLP-MABSA) and trained it on three types of task-specific pre-training tasks. Yu et al. (2022) introduced a multi-task learning architecture to capture both coarse-grained and fine-grained image-target matching relations.

## 2.2 Instruction Learning

Recently, we have witnessed remarkable progress in enhancing the cross-task generalization of models via instructions, such as FLAN (Wei et al., 2022), PromptSource (Sanh et al., 2022), Instruct-GPT (Ouyang et al., 2022) and Tk-Instruct (Wang et al., 2022b). Furthermore, a myriad of approaches utilizing instructions to handle domain-specific tasks have been proposed (Kuznia et al., 2022; Luo et al., 2022; Varia et al., 2022; Scaria et al., 2023; Wang et al., 2022a).

## 3 Method

### 3.1 Task Formalization

Given a set of multimodal samples $\Gamma$, a sample $\gamma \in \Gamma$ is a tuple $(S, V, T, y)$, where $S = \{w_1, \ldots, w_i, \ldots, w_{i+|T|-1}, \ldots, w_{|S|}\}$ is a $|S|$-word sentence, $V$ is an image, $T = \{w_i, \ldots, w_{i+|T|-1}\} \subseteq S$ denotes a $|T|$-word target term, and $y \in \{positive, negative, neutral\}$ indicates the sentiment polarity towards the target term. Our goal is to learn a function to predict the sentiment polarity $y$.

### 3.2 Overview

As shown in Figure 2, our model consists of four modules: *TSEI Extraction and Refinement* module, *Image Scene Caption Generation* module, *TANPs Extraction and Alignment* module, and *Instruction Learning with Prompts* module. Given a multimodal input $(S, V, T)$ where $S$ denotes the sentence and $V$ indicates the image, in the first three modules, we extract TSEI, image scene caption and TANPs from $V$ in natural language format respectively, which preserve both coarse-grained and fine-grained descriptions of $V$ and thus harvest a sufficient comprehension of visual semantic information. In the fourth module, we integrate the extracted TSEI, image scene caption and TANPs with $S$ to manually design instruction prompts for instruction-tuning Tk-Instruct, which generates the sentiment prediction in the output. We will demonstrate the model details in the following subsections.

## 3.3 TSEI Extraction and Refinement

In this module, we extract TSEI from the image $V$ and refine them to reduce noises.

**Extraction with Maximum Area Strategy.** We utilize a well-performing OCR model Easy-OCR[2] to recognize TSEI from the image $V$ and select the primary TSEI from the region with the maximum area. The distribution of TSEI in $V$ can be divided into multiple bounding boxes. TSEI from different bounding boxes are usually semantically incoherent if they are spliced together. Besides, TSEI from small bounding boxes are often too brief to provide valuable information, and instead add noises. Therefore, we concentrate on the primary content and select the recognized TSEI from the bounding box with the maximum area:

$$B_i = (X_{i,0}, X_{i,1}, Y_{i,0}, Y_{i,1}, R_i), i = 1, \ldots, I \quad (1)$$

$$\mathcal{A}_i = |X_{i,1} - X_{i,0}| * |Y_{i,1} - Y_{i,0}| \quad (2)$$

$$O' = R_j, \text{where} \quad j = \arg \max_{i=1,\ldots,I} \mathcal{A}_i \quad (3)$$

where $B_i$ denotes the $i_{th}$ bounding box; $X_{i,0}$, $X_{i,1}$, $Y_{i,0}$ and $Y_{i,1}$ are the borderline coordinates of the bounding box; $R_i$ is the recognized TSEI; $\mathcal{A}_i$ denotes the area of the bounding box; and $O'$ indicates the selected TSEI.

**Spelling Correction.** Due to the limitation of the image quality and EasyOCR, there will be some errors in $O'$ such as misrecognition of characters and inappropriate spaces. Thus, we employ Symmetric Delete spelling correction algorithm[3] to correct and modify $O'$:

$$O'' = \text{SymSpell}(O') \quad (4)$$

**Grammar Filtering.** We feed $O''$ into the RoBERTa fine-tuned on the CoLA dataset (Warstadt et al., 2018) to judge the grammatical acceptability of $O''$. We filter out those with ungrammatical confidence above the threshold $\theta_1 = 0.9$. The reserved $O''$ is the resulted TSEI.

$$H_{O''} = \text{RoBERTa}(O'') \quad (5)$$

$$J = \text{Softmax}(W H_{O''}^{[CLS]} + b) \quad (6)$$

where $W \in \mathbb{R}^{768 \times 2}$ and $b \in \mathbb{R}^2$ are parameters learned in fine-tuning; and $H_{O''}^{[CLS]} \in \mathbb{R}^{768}$ is the final hidden state of the $[CLS]$ token.

---

[2]https://www.jaided.ai/easyocr/
[3]https://github.com/wolfgarbe/SymSpell

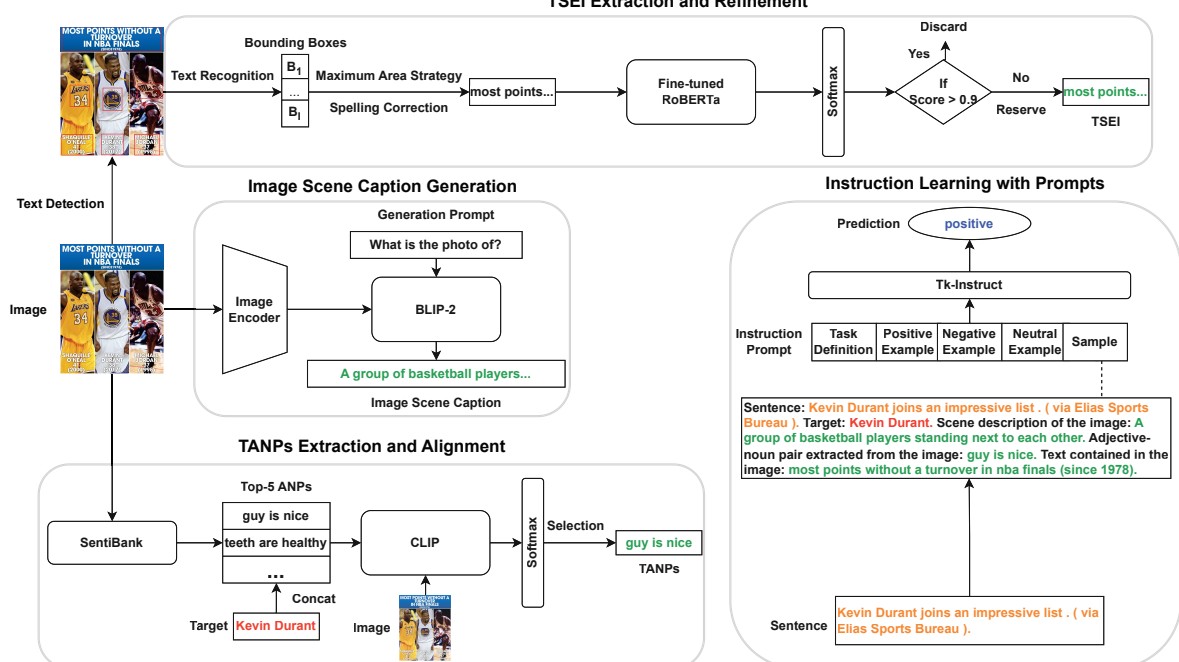

Figure 2: The architecture of our VEMP.

## 3.4 Image Scene Caption Generation

We apply a state-of-the-art vision-language pre-trained model BLIP-2 (Li et al., 2023) to generate textual image scene caption of the image $V$ with the following procedures.

First, we encode the image $V$ with a frozen image encoder to get the embeddings:

$$H_V = \text{Image-Encoder}(V) \qquad (7)$$

Then, we interact a set of learnable query embeddings as soft prompts with $H_V$ to extract language-informative visual representation $H_V'$ in a querying transformer:

$$H_V' = \text{FF}(\text{Cross-ATT}(\text{Self-ATT}(Q), H_V)) \qquad (8)$$

where $Q$ denotes the query embeddings; Self-ATT denotes self-attention layer; Cross-ATT denotes cross-attention layer; and FF denotes feed forward layer.

Next, a fully-connected (FC) layer projects $H_V'$ into the same hidden dimension of a large language model (LLM):

$$H_V'' = \text{FC}(H_V') \qquad (9)$$

Finally, we pass the generation prompt $P$ along with $H_V''$ to the LLM to generate the image scene caption:

$$C = \text{LLM-Decoder}(H_V'', P) \qquad (10)$$

## 3.5 TANPs Extraction and Alignment

We extract ANPs from the image $V$ and conduct alignment with the target $T$ to obtain TANPs.

**ANPs Extraction.** We follow KEF to use SentiBank toolkit[4] to extract ANPs with the Top-5 confidence. And we rewrite these ANPs as "Noun is/are Adjective".

**Alignment with Target.** We utilize the model CLIP (Radford et al., 2021) to evaluate the relevance between ANPs and the target for alignment. We concatenate ANPs with the target and feed them into the text encoder of CLIP, while input the image into the image encoder of CLIP. Benefiting from pre-training on a large-scale dataset of text-image pairs, CLIP is able to work out the similarity score, which is subsequently passed to a softmax layer to calculate the matching probability. We select ANPs with matching probability exceeding the threshold $\theta_2 = 0.5$.

$$M = \text{Softmax}(\text{CLIP}(\text{concat}(A, T), V)) \qquad (11)$$

$$A_{T \to A} = A_i, \text{where} \quad M_i > \theta_2 \quad i = 1, \dots, 5 \qquad (12)$$

where $A$ denotes ANPs; $M \in \mathbb{R}^5$ denotes the matching probability; and $A_{T \to A}$ indicates the resulted TANPs.

---

[4]ee.columbia.edu/ln/dvmm/vso/download/sentibank.html

| | Twitter-2015 | | | | | | Twitter-2017 | | | | | |
|-------|------|------|-----|-------|---------|-------|------|-----|------|-------|---------|-------|
| Split | Pos | Neu | Neg | Total | Targets | Len | Pos | Neu | Neg | Total | Targets | Len |
| Train | 928 | 1883 | 368 | 3179 | 1.34 | 16.72 | 1508 | 416 | 1638 | 3562 | 1.41 | 16.21 |
| Dev. | 303 | 679 | 149 | 1122 | 1.33 | 16.74 | 515 | 144 | 517 | 1176 | 1.43 | 16.37 |
| Test | 317 | 607 | 113 | 1037 | 1.35 | 17.05 | 493 | 168 | 573 | 1234 | 1.45 | 16.38 |

Table 1: The statistics of two benchmark datasets. Pos: Positive, Neu: Neutral, Neg: Negative, Len: Length.

| | Twitter-TSEI | | | | | |
|-------|------|------|-----|-------|---------|-------|
| Split | Pos | Neu | Neg | Total | Targets | Len |
| Train | 1114 | 1568 | 347 | 3029 | 1.38 | 16.42 |
| Dev. | 393 | 552 | 131 | 1076 | 1.39 | 16.53 |
| Test | 358 | 527 | 133 | 1018 | 1.41 | 16.67 |

Table 2: The statistics of Twitter-TSEI dataset. Pos: Positive, Neu: Neutral, Neg: Negative, Len: Length.

### 3.6 Instruction Learning with Prompts

In this module, we aim to fuse the sentence $S$ with the visual elements mined from the image $V$ for modeling. To this end, we leverage the sentence $S$, the target $T$, TSEI, TANPs and image scene caption to manually design instruction prompts, and then apply these instruction prompts to instruction-tune Tk-Instruct for generating the sentiment prediction.

Appropriate prompts are essential for stimulating the learning potential of language models (LMs). Empirically, based on the prior success (Scaria et al., 2023), we make an effort to ensure the instruction prompts are concise, explicit, and easily comprehensible. Specifically, each of our instruction prompts comprises a task definition, three examples of "positive", "negative" and "neutral" sentiment labels respectively, and a to-be-classified sample. The format of every example or sample is as "Sentence: $S$. Target: $T$. Scene description of the image: $C$. Adjective-noun pair extracted from the image: $A_{T \rightarrow A}$. Text contained in the image: $O''$", where $C$, $A_{T \rightarrow A}$ and $O''$ denote image scene caption, TANPs and TSEI respectively. We employ specific criteria for the selection of our few-shot examples. Firstly, "positive", "negative", and "neutral" sentiment samples should all be included. Given the constraint imposed by Tk-Instruct's maximum input tokens limit and the lengthy nature of our examples, we opt for 1-shot examples for each sentiment category. Secondly, since our classification task focuses on fine-grained targets, we have at least two examples that share the same input but with different targets and sentiment labels. Lastly, we choose samples with acknowledged and uncontroversial sentiment labels. The more detailed instruction prompts example is shown in Appendix A.

We use the following loss for instruction-tuning:

$$\mathcal{L} = -\frac{1}{|\Gamma|} \sum_{i=1}^{|\Gamma|} \log p_\theta(y^i|x^i) \tag{13}$$

where $x^i$ denotes the input instruction prompts sequence; $\theta$ is the model parameters; and $y^i$ indicates the output sentiment prediction token.

## 4 Experiment

To validate the advantage of our method, we compare it with state-of-the-art models and GPT-4. And we perform ablation study and case study to verify the effectiveness of each module of our method.

### 4.1 Experimental Setup

**Datasets.** We evaluate our method on two benchmark datasets Twitter-2015 and Twitter-2017. Besides, we select samples with TSEI from Twitter-2015 and Twitter-2017 to build a new dataset Twitter-TSEI to check the effect of our method on more complex dataset. The statistics of Twitter-2015, Twitter-2017 and Twitter-TSEI are shown in Table 1 and Table 2.

**Implementation Details.** We employ the model Tk-Instruct-base-def-pos as backbone, and we set the batch size, the learning rate and the training epochs as 16, 5e-5 and 30 respectively. All instruction-tuning experiments are based on PyTorch with an NVIDIA GeForce RTX 3090 GPU.

**Evaluation Metrics** We use Accuracy (Acc) and Macro-F1 as evaluation metrics, and report the average of 5 independent training runs as results.

| Method | Twitter-2015 | | Twitter-2017 | |
|---|---|---|---|---|
| | Acc | Macro-F1 | Acc | Macro-F1 |
| Visual Only | | | | |
| Res-Target | 59.88 | 46.48 | 58.59 | 53.98 |
| Text Only | | | | |
| AE-LSTM | 70.30 | 63.43 | 61.67 | 57.97 |
| MGAN | 71.17 | 64.21 | 64.75 | 61.46 |
| BERT | 74.15 | 68.86 | 68.15 | 65.23 |
| Tk-Instruct | 77.53 | 71.88 | 71.07 | 69.66 |
| Text and Visual | | | | |
| MIMN | 71.84 | 65.69 | 65.88 | 62.99 |
| ESAFN | 73.38 | 67.37 | 67.83 | 64.22 |
| TomBERT | 77.15 | 71.75 | 70.50 | 68.04 |
| EF-CapTrBERT | 78.01 | 73.25 | 69.77 | 68.42 |
| FITE | 78.49 | 73.90 | 70.90 | 68.70 |
| KEF-TomBERT | 78.68 | 73.75 | 72.12 | 69.96 |
| VLP-MABSA | 78.60 | 73.80 | **73.80** | 71.80 |
| ITM | 78.27 | 74.19 | 72.61 | 71.97 |
| Our Method | | | | |
| VEMP | **78.88** | **75.09** | 73.01 | **72.42** |

Table 3: Experimental results for TMSC.

| Method | Twitter-TSEI | |
|---|---|---|
| | Acc | Macro-F1 |
| Text Only | | |
| Tk-Instruct | 67.68 | 64.23 |
| Text and Visual | | |
| ITM | 68.86 | 65.59 |
| VEMP | 71.72(+2.86%) | 68.61(+3.02%) |

Table 4: Experimental results on Twitter-TSEI dataset.

## 4.2 Compared Baselines

We compare our method with three kinds of baselines as follows:

**Visual-Only**. Res-Target, which concatenates the target embeddings and the visual features extracted by ResNet (He et al., 2016) for classification.

**Text-Only**. (1) AE-LSTM (Wang et al., 2016), an attention-based LSTM which utilizes attention mechanism to concentrate on the aspect in a sentence. (2) MGAN (Fan et al., 2018), a Multi-Grained Attention Network which aims to alleviate the information loss of coarse-grained attention mechanism. (3) BERT (Devlin et al., 2019), a bidirectional pre-trained language model which has context understanding abilities. (4) Tk-Instruct (Wang et al., 2022b), a generative pre-

trained language model which is equipped with strong learning capabilities to unseen tasks due to rigorous training following instructions.

**Text and Visual**. (1) MIMN (Xu et al., 2019), a Multi-Interactive Memory Network which learns both the cross-modality and the single-modality interactions. (2) ESAFN (Yu et al., 2019), an Entity-Sensitive Attention and Fusion Network. (3) TomBERT (Yu and Jiang, 2019), which applys BERT and attention mechanism to model intra-modality dynamics and inter-modality dynamics. (4) EF-CapTrBERT (Khan and Fu, 2021), which translates images into auxiliary sentences and then fuse with text-modality through a pre-trained language model. (5) FITE (Yang et al., 2022), which translates facial expressions in images into emotional texts for fusion with text-modality. (6) KEF-TomBERT (Zhao et al., 2022), which combines a knowledge-enhanced framework with TomBERT to improve the capability of visual attention and sentiment prediction. (7) VLP-MABSA (Ling et al., 2022), which adopts task-specific vision-language pre-training. (8) ITM (Yu et al., 2022), a multi-task learning architecture which captures both coarse-grained and fine-grained image-target matching relations.

## 4.3 Experimental Results and Analysis

In the subsection, we compare our method with other baselines on Twitter-2015, Twitter-2017 and Twitter-TSEI, and provide empirical analysis.

Based on the experimental results on Twitter-2015 and Twitter-2017 datasets shown in Table 3, our observations are as follows: (1) Compared to EF-CapTrBERT and FITE, which also translate visual modality into text modality for cross-modal fusion, our VEMP outweighs EF-CapTrBERT by 1.84% and 4.00%, and surpasses FITE by 1.19% and 3.72%, on the Macro-F1 score on the Twitter-2015 dataset and the Twitter-2017 dataset respectively. Besides, our VEMP also outperforms the state-of-the-art model ITM. These advancements corroborate the remarkable superiority of our method. (2) Tk-Instruct outperforms all other text-only methods and even surpasses multimodal method TomBERT, which demonstrates its excellent learning capabilities under instructions. (3) Our VEMP improves text-only Tk-Instruct on the Macro-F1 score by 3.21% on the Twitter-2015 dataset, and 2.76% on the Twitter-2017 dataset, which demonstrate the effectiveness of our VEMP

| Method | Twitter-2015 | | Twitter-2017 | |
|---|---|---|---|---|
| | Acc | Macro-F1 | Acc | Macro-F1 |
| VEMP | **78.88** | **75.09** | **73.01** | **72.42** |
| w/o TSEI | 77.82 | 73.70 | 71.56 | 70.77 |
| w/o Image Scene Caption | 78.40 | 74.38 | 72.04 | 70.88 |
| w/o TANPs | 78.21 | 74.86 | 71.56 | 70.20 |

Table 5: Ablation study of the three visual elements mining modules in our VEMP. "w/o" indicates the removal of the corresponding module.

| Method | Twitter-2015 | | Twitter-2017 | |
|---|---|---|---|---|
| | Acc | Macro-F1 | Acc | Macro-F1 |
| VEMP | **78.88** | **75.09** | **73.01** | **72.42** |
| w/o Spelling Correction | 77.92 | 73.81 | 71.56 | 70.79 |
| w/o Grammar Filtering | 78.68 | 74.61 | 71.31 | 70.11 |
| w/o Spelling Correction & Grammar Filtering | 78.11 | 74.34 | 71.72 | 70.09 |
| w/o Alignment with Target | 78.01 | 74.23 | 71.88 | 70.59 |

Table 6: Ablation study of detailed units in our VEMP. "w/o" indicates the removal of the unit.

in leveraging visual modality for training models.

We also report the experimental results on Twitter-TSEI dataset in Table 4. We can find that VEMP achieves a more significant improvement over ITM than on Twitter-2015 and Twitter-2017. It is reasonable because ITM ignores TSEI while we attach importance to them in our VEMP. Moreover, VEMP still outperforms the text-only Tk-Instruct since VEMP can effectively exploit visual information.

## 4.4 Ablation Study

**Three Modules for Visual Elements Mining.** To investigate the impact of *TSEI Extraction and Refinement* module, *Image Scene Caption Generation* module and *TANPs Extraction and Alignment* module in our model, we remove them separately and observe the resulting consequences.

As depicted in Table 5, we can find that: (1) The performance declines significantly after removing the *TSEI Extraction and Refinement* module. Specifically, the Accuracy and Macro-F1 score drop 1.06% and 1.39% on the Twitter-2015 dataset respectively, and the Accuracy and Macro-F1 score drop 1.45% and 1.65% on the Twitter-2017 dataset respectively. This indicates that the *TSEI Extraction and Refinement* module can effectively yield TSEI as beneficial information for TMSC. (2) Without the *Image Scene Caption Generation* module, the Accuracy and Macro-F1 score decline about 0.5% on the Twitter-2015 dataset and about 1% on the Twitter-2017 dataset, which demonstrates the importance of the *Image Scene Caption Generation* module in translating the im-

| Method | Twitter-2015 | | Twitter-2017 | |
|---|---|---|---|---|
| | Acc | Macro-F1 | Acc | Macro-F1 |
| Text Only | | | | |
| GPT-4(zero-shot) | 64.61 | 59.35 | 59.24 | 57.60 |
| GPT-4(few-shot) | 65.19 | 59.88 | 59.97 | 58.73 |
| GPT-3(fine-tuning) | 76.66 | 71.02 | 68.23 | 66.22 |
| Tk-Instruct(instruction-tuning) | 77.53 | 71.88 | 71.07 | 69.66 |
| Text and Visual | | | | |
| VEMP$_{GPT-4}$(zero-shot) | 65.00 | 59.95 | 59.56 | 58.29 |
| VEMP$_{GPT-4}$(few-shot) | 65.77 | 61.19 | 60.62 | 59.48 |
| VEMP$_{GPT-3}$(fine-tuning) | 77.43 | 71.86 | 68.72 | 66.83 |
| VEMP$_{Tk-Instruct}$(instruction-tuning) | **78.88** | **75.09** | **73.01** | **72.42** |

Table 7: Experimental results of different learning paradigms. For example, VEMP$_{GPT-4}$(zero-shot) indicates that we utilize VEMP to mine visual elements for zero-shot learning on GPT-4.

age into textual scene outline for cross-modal fusion. (3) Eliminating the *TANPs Extraction and Alignment* module results in performance degradation as well. In particular, the Accuracy declines 1.45% and the Macor-F1 score drops 2.22% on the Twitter-2017 dataset. This illustrates the *TANPs Extraction and Alignment* module's contribution to distill TANPs for emotional guidance. (4) Remarkably, we notice that removing the *TSEI Extraction and Refinement* module leads to the most intense performance decay, which validates the rationality of our motivation to leverage TSEI for TMSC.

**Detailed Units for Visual Elements Mining.** We implement further ablation study to check the influence of more detailed units in our VEMP. We remove the *Spelling Correction* unit, the *Grammar Filtering* unit, both *Spelling Correction* and *Grammar Filtering* units, and the *Alignment with Target* unit respectively.

Based on the experimental results in Table 6, we can observe that the performance downgrades on the Twitter-2015 dataset and the Twitter-2017 dataset after removing the *Spelling Correction* unit, the *Grammar Filtering* unit or the both of them, which confirms their indispensable impacts on denoising the extracted TSEI. Besides, directly applying the ANPs without alignment with the target also results in performance decline since the ANPs irrelevant to the target add noises.

***Instruction Learning with Prompts* Module.** To evaluate the effectiveness of *Instruction Learning with Prompts* module, we replace it with other typical learning paradigms, such as fine-tuning on GPT-3 (Brown et al., 2020), zero-shot learning on GPT-4 and few-shot learning on GPT-4. For fine-tuning GPT-3, We adopt the base model ada

| Image | 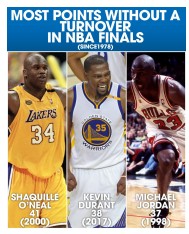 | 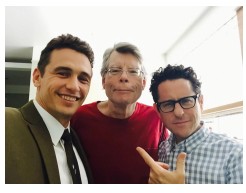 | |
|---|---|---|---|
| Sentence | (a) [Cleveland Cavaliers]_negative , Marty Huggins said it best ! # Warriors # Cavs # sweep # nbafinals. | (b) [Kevin Durant]_positive joins an impressive list . ( via Elias Sports Bureau ). | (c) Me , along with a couple of friends - - [James Franco]_positive and JJ Abrams - - on the set of 11 / 22 / 63 . Everything looks cool . None. |
| TSEI | because it's a mess. | most points without a turnover in nba finals (since1978). | |
| Image Scene Caption | A man in a suit and tie giving a speech. | A group of basketball players standing next to each other. | Three men posing for a picture in a room. |
| TANPs | man is angry. | guy is nice. | family is happy. |
| Tk-Instruct | positive ✗ | positive ✓ | neutral ✗ |
| ITM | positive ✗ | negative ✗ | positive ✓ |
| VEMP | negative ✓ | positive ✓ | positive ✓ |

Table 8: Predictions of text-only Tk-Instruct, ITM and VEMP on three test samples.

and set the batch size, the learning rate and the training epochs as 8, 0.05 and 4 respectively. All fine-tuning experiments are implemented on OpenAI servers remotely via OpenAI API. For zero-shot and few-shot learning on GPT-4[5], we design four kinds of prompts tailored for text-only zero-shot setting, text-only few-shot setting, multimodal zero-shot setting and multimodal few-shot setting respectively, which are adapted from the report (Zhang et al., 2023) that utilized large language models (LLMs) for sentiment analysis. To ensure a fair comparison, we employ the same few-shot examples as instruction learning on Tk-Instruct. Specific prompt examples are shown in Appendix A.

Table 7 depicts the experimental results of different learning paradigms. We can observe that: (1) GPT-4 with zero-shot or few-shot setting performs poorly on TMSC. We conjecture the reasons are that TMSC concerns the fine-grained target and sentences in tweets are informal, which increase the complexity of the task so that zero-shot and few-shot learning are incompetent. (2) Instruction-tuning Tk-Instruct surpasses other learning paradigms by a wide margin on both text-only and multimodal data, which demonstrates the effectiveness of *Instruction Learning with Prompts* module. Furthermore, we can conclude that instruction-tuning on language models is competitive and promising in the era of LLMs.

## 4.5 Case Study

To understand the advantage of our method intuitively, we present the predictions of text-only Tk-Instruct, ITM and VEMP on three test samples, as shown in Table 8. For sample (a), with the aid of the TSEI "because it's a mess" and the TANPs "man is angry", our VEMP made the correct prediction whereas other models fell into errors. For sample (b), ITM made a wrong prediction since it just focused on the neutral visual content excluding TSEI. In contrast, VEMP exploited the TSEI "most points without a turnover in nba finals (since1978)" which conveyed a positive emotional tendency and thus gave the correct prediction. For sample (c), despite without TSEI in the image, VEMP still predicted the sentiment polarity correctly, benefiting from the TANPs "family is happy".

## 5 Conclusion

In this paper, we propose a Visual Elements Mining as Prompts (VEMP) method for Target-Oriented Multimodal Sentiment Classification (TMSC). In order to describe the image with sufficient details and emotional characterization, we extract Text Symbols Embedded in the Image (TSEI) and Target-aware Adjective-Noun Pairs (TANPs) as supplements to image scene caption. And then we leverage them to construct instruction prompts to instruction-tune Tk-Instruct for sentiment prediction. Experimental results show that our method outperforms state-of-the-art ap-

---

[5]The experiments are based on the August 23 version of GPT-4. It is important to note that future updates to the model may potentially impact the results presented in this paper.

proaches and GPT-4. Our approach is universally applicable to all types of images, thus we intend to extend it to other multimodal tasks in the future.

## Limitations

Our method has a notable limitation that it heavily relies on the precision of visual elements mining results. Despite employing state-of-the-art computer vision models and carefully designing post-processing procedures, there still exists a certain level of errors and noises in the extracted results. Another limitation is that we depend on the text understanding capabilities of Tk-Instruct and instruction-tuning operation to align targets with sentences. In the future, we should explore more effective alignment strategies. In addition, the Tk-Instruct we have chosen may not be considered cutting-edge. Moving forward, we plan to instruction-tune more advanced language models such as GPT-3.5 and GPT-4, provided that they offer fine-tuning accesses.

## Ethics Statement

Our work employs uniform code for all images to extract TSEI, TANPs and image scene caption, ensuring objectivity and avoiding bias. Furthermore, it utilizes two publicly available Twitter datasets while adhering to Twitter's data policy.

## Acknowledgements

We are greatly appreciative of the anonymous reviewers for their valuable feedback and suggestions.

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

## A Appendix

| | |
|---|---|
| Definition | The output will be 'positive' if the target identified in the sentence contains a positive sentiment. If the sentiment of the identified target in the input is negative the answer will be 'negative'. Otherwise, the output should be 'neutral'. |
| Positive example | Input: Sentence: RT @ BarackObama : # ClimateChangeIsReal . So are the actions $T$ is taking to tackle it . Target: President Obama. Scene description of the image: A man in a white shirt and tie standing in front of trees. Adjective-noun pair extracted from the image: hand is helping. Text contained in the image: the strongest action ever taken by an american president to tackle climate change. Output: positive |
| Negative example | Input: Sentence: Former Bridgecorp boss $T$ will be released from jail next month . Target: Rod Petricevic. Scene description of the image: A man in a suit and tie standing in a room. Adjective-noun pair extracted from the image: performance is poor. Text contained in the image: None. Output: negative |
| Neutral example | Input: Sentence: Former $T$ boss Rod Petricevic will be released from jail next month . Target: Bridgecorp. Scene description of the image: A man in a suit and tie standing in a room. Adjective-noun pair extracted from the image: performance is poor. Text contained in the image: None. Output: neutral |
| Input Sample | Now complete the following example- Input: Sentence: $T$ joins an impressive list . ( via Elias Sports Bureau ). Target: Kevin Durant. Scene description of the image: A group of basketball players standing next to each other. Adjective-noun pair extracted from the image: guy is nice. Text contained in the image: most points without a turnover in nba finals (since 1978). Output: |
| Label | positive |

Table 9: An instruction prompt example for instruction learning in our VEMP.

| | |
|---|---|
| Definition | Please perform Target-Oriented Sentiment Classification task. Every sample is a tweet from twitter which contains a sentence and a target in the sentence. The input provides a sample's sentence and target. Given the input, you should identify the sentiment towards the target and assign a sentiment label from ['positive', 'negative', 'neutral']. Return label only without any other text. |
| Input Sample | Input: Sentence: $T$ joins an impressive list . ( via Elias Sports Bureau ). Target: Kevin Durant. |

Table 10: A prompt example for text-only zero-shot setting on GPT-4.

| | |
|---|---|
| Definition | Please perform Target-Oriented Sentiment Classification task. Every sample is a tweet from twitter which contains a sentence and a target in the sentence. The input provides a sample's sentence and target. Given the input, you should identify the sentiment towards the target and assign a sentiment label from ['positive', 'negative', 'neutral']. Return label only without any other text. |
| Examples | Input: Sentence: RT @ BarackObama : # ClimateChangeIsReal . So are the actions President Obama is taking to tackle it . Target: President Obama. Label:positive
Input: Sentence: Former Bridgecorp boss Rod Petricevic will be released from jail next month . Target: Rod Petricevic. Label:negative
Input: Sentence: Former Bridgecorp boss Rod Petricevic will be released from jail next month . Target: Bridgecorp. Label:neutral |
| Input Sample | Input: Sentence: $T$ joins an impressive list . ( via Elias Sports Bureau ). Target: Kevin Durant. |

Table 11: A prompt example for text-only few-shot setting on GPT-4.

| | |
|---|---|
| Definition | Please perform Target-Oriented Multimodal Sentiment Classification task. Every sample is a tweet from twitter which contains a sentence, a target in the sentence and an image. The input provides a sample's sentence, target, scene description of the image, adjective-noun pair extracted from the image and text contained in the image. Given the input, you should identify the sentiment towards the target and assign a sentiment label from ['positive', 'negative', 'neutral']. Return label only without any other text. |
| Input Sample | Input: Sentence: $T$ joins an impressive list . ( via Elias Sports Bureau ). Target: Kevin Durant. Scene description of the image: A group of basketball players standing next to each other. Adjective-noun pair extracted from the image: guy is nice. Text contained in the image: most points without a turnover in nba finals (since 1978). |

Table 12: A prompt example for multimodal zero-shot setting on GPT-4.

| | |
|---|---|
| Definition | Please perform Target-Oriented Multimodal Sentiment Classification task. Every sample is a tweet from twitter which contains a sentence, a target in the sentence and an image. The input provides a sample's sentence, target, scene description of the image, adjective-noun pair extracted from the image and text contained in the image. Given the input, you should identify the sentiment towards the target and assign a sentiment label from ['positive', 'negative', 'neutral']. Return label only without any other text. |
| Examples | Input: Sentence: RT @ BarackObama : # ClimateChangeIsReal . So are the actions President Obama is taking to tackle it . Target: President Obama. Scene description of the image: A man in a white shirt and tie standing in front of trees. Adjective-noun pair extracted from the image: hand is helping. Text contained in the image: the strongest action ever taken by an american president to tackle climate change. Label:positive 
 Input: Sentence: Former Bridgecorp boss Rod Petricevic will be released from jail next month . Target: Rod Petricevic. Scene description of the image : A man in a suit and tie standing in a room. Adjective-noun pair extracted from the image: performance is poor. Text contained in the image: None. Label:negative 
 Input: Sentence: Former Bridgecorp boss Rod Petricevic will be released from jail next month . Target: Bridgecorp. Scene description of the image: A man in a suit and tie standing in a room. Adjective-noun pair extracted from the image: performance is poor. Text contained in the image: None. Label:neutral |
| Input Sample | Input: Sentence: $T$ joins an impressive list . ( via Elias Sports Bureau ). Target: Kevin Durant. Scene description of the image: A group of basketball players standing next to each other. Adjective-noun pair extracted from the image: guy is nice. Text contained in the image: most points without a turnover in nba finals (since 1978). |

Table 13: A prompt example for multimodal few-shot setting on GPT-4.