# OpenReview forum: "Visual Elements Mining as Prompts for Instruction Learning for Target-Oriented Multimodal Sentiment Classification"
_EMNLP/2023/Conference — EMNLP 2023 Findings_

### Official Review · Reviewer_HyRf · 2023-08-01

**Soundness:** 3

**Excitement:**

3: Ambivalent: It has merits (e.g., it reports state-of-the-art results, the idea is nice), but there are key weaknesses (e.g., it describes incremental work), and it can significantly benefit from another round of revision. However, I won't object to accepting it if my co-reviewers champion it.

**Paper Topic And Main Contributions:**

The manuscript focuses on the Target-oriented multimodal sentiment classification task and proposes a Visual Elements Mining as Prompts model to address the task. The main contributions are the consideration of text symbols in images and adjective-noun pairs in forming instruction prompts and state-of-the-art results on two datasets.

**Questions For The Authors:**

See Reasons to reject.

**Reasons To Accept:**

1. The considerations of text symbols in images and the adjective-noun pairs.
2. Instruction tuning

**Reasons To Reject:**

1. Less exciting model framework: basically every module adopts an existing model so the novelty is limited. The reasons of choosing these models are also not clear. Only the instruction prompt and instruction tuning look interesting, but the fine-tuning process is also mediocre.

2. Is providing one example in the prompt enough? Will increasing this number be helpful? Will the order of presenting information affect the instruction learning results?

3. How sensitive are the model performance to the choice of selected models in each module? for example, what if the model CLIP is replaced by another model, will the model performance be affected much?

**Reproducibility:**

3: Could reproduce the results with some difficulty. The settings of parameters are underspecified or subjectively determined; the training/evaluation data are not widely available.

**Reviewer Confidence:**

3: Pretty sure, but there's a chance I missed something. Although I have a good feel for this area in general, I did not carefully check the paper's details, e.g., the math, experimental design, or novelty.

---

> ### Author Rebuttal · Authors · 2023-08-27
>
> We greatly appreciate your diligent review and feedback. The following is our responses to the questions you raised.
>
> Q: Less exciting model framework: basically every module adopts an existing model so the novelty is limited. The reasons of choosing these models are also not clear. Only the instruction prompt and instruction tuning look interesting, but the fine-tuning process is also mediocre.
>
> A: The reason to choose these models: They were the state-of-the-art and best-suited models to meet our requirements for extracting visual elements to the best of our knowledge. **For example, to generate more accurate image caption, we tried and compared multiple models: Caption Transformer (Khan and Fu, 2021, as cited in line 47), BLIP, and BLIP-2. And we selected BLIP-2 since it exhibited the best caption generation performance**.
>
> Q: Is providing one example in the prompt enough? Will increasing this number be helpful? Will the order of presenting information affect the instruction learning results?
>
> A: **We used 3-shot examples rather than one example (as shown in Appendix Table 9)**.    \
> **The reason to select only 3-shot examples**: Due to the maximum input tokens constraint of Tk-Instruct and the long length of our examples, the number was limited.   \
> In the process of experiments, we did not observe that the order of presenting the 3-shot examples affected the results significantly.
>
> Q: How sensitive are the model performance to the choice of selected models in each module? for example, what if the model CLIP is replaced by another model, will the model performance be affected much?
>
> A: The selected models were the state-of-the-art and best-suited ones to meet our requirements for extracting visual elements to the best of our knowledge. If they are replaced with other inferior models, the extraction results may be worse, potentially leading to performance decline. For example, to generate more accurate image caption, we tried and compared multiple models: Caption Transformer (Khan and Fu, 2021, as cited in line 47), BLIP, and BLIP-2. And BLIP-2 exhibited relatively better caption generation performance and entire model performance than the others.

---

### Official Review · Reviewer_BJVD · 2023-08-05

**Soundness:** 3

**Ethical Concerns:**

Yes

**Excitement:**

3: Ambivalent: It has merits (e.g., it reports state-of-the-art results, the idea is nice), but there are key weaknesses (e.g., it describes incremental work), and it can significantly benefit from another round of revision. However, I won't object to accepting it if my co-reviewers champion it.

**Justification For Ethical Concerns:**

The ethics statement of the paper seemed lacking and could have discussed issues around how their approach is influenced by the biased captioning of image and detection of adjective-noun pairs in images. Additionally, the authors do not discuss the data availability and related ethics. They use two existing Twitter datasets and it is unclear if the datasets contain just the Tweet IDs (if that is the case, the data may no longer be available for reproducibility studies or follow-up work).

**Paper Topic And Main Contributions:**

The paper proposes an approach for classifying the sentiment of multimodal data (visual + text) data towards a target. This is done by extracting text in the image (OCR), and target-aware adjective-noun pairs (TANPs), along with the image caption. The authors argue that including the first two attributes (i.e., OCR and TANPs) along witht the caption to construct an instruction prompt for instruct-tuning a language-only model, would lead to better performance than existing baseliens. To this end, their experiments demonstrate better performance on two-datasets.

**Questions For The Authors:**

A: My major concern with the paper is that many details that are critical to ensure the validty of comparisons with the baselines are not discussed in the paper. I have enumerated these below. However, as a consequence of these details missing from the paper, I am not completely convinced of the approach being superior to the considered baselines.
- (i): how were the baseline models trained and was there any hyperparameter search while training them?
- (ii): How were the hyperparameters for the proposed approach (descibed in Section 4.1) set?
- (iii): What is the 'ada' model that the authors used for GPT-3 (there are several variants)? Similarly, which ChatGPT model did the authors use (gpt-3.5; gpt-4; which dates? as the models were updated frequently?)
- (iv): How did the authors choose the few-shot examples provided to ChatGPT? Were there any evaluations done to quantify the sensitivity of results to the few-shot examples being used in-context?

B. I am also unsure about the validity of the approach that the authors adopt to extract text from images. It is possible that in some cases the text in the image is split between two different blocks (for instance, in Figure 1, one half of the text is in the top of the image and the other half is in the bottom). Authors mention that they select the bounding box with the maximum area? Does this mean that the authors only consider only half of the text within the image? This would provide incomplete context to the later stages of the approach.

C. The authors find that Tk-instruct perform better than ChatGPT (not sure which specific model, as it is not mentioned) but that contary to what has been observed in some recent work (Labrak et al., 2023: https://arxiv.org/pdf/2307.12114.pdf). This makes me wonder if the authors adopted approaches like Chain of Thought prompting (Wet el al., 2022: https://arxiv.org/pdf/2201.11903.pdf) to accurately assess the capability of ChatGPT as a baseline. The prompts provided in the appendix do not contain such strategies.

**Reasons To Accept:**

-  The idea of extracting the additional information (OCR and TANPs) from the image to boost target-aware sentiment classification is an interesting one
- The authors compare against several text-only and multimodal baselines, as well as one image-only baseline. Overall, the range of experiments in the paper is good

**Reasons To Reject:**

- While there are several experiments in the paper, some important details are missing in many places
- Several design choices in the paper are not explained or mentioned without any proper justification
- The paper uses many acronyms, which gets a bit confusing and often gets in the way of comprehension
I expand on these points in my questions to the authors.

**Reproducibility:**

3: Could reproduce the results with some difficulty. The settings of parameters are underspecified or subjectively determined; the training/evaluation data are not widely available.

**Reviewer Confidence:**

4: Quite sure. I tried to check the important points carefully. It's unlikely, though conceivable, that I missed something that should affect my ratings.

**Typos Grammar Style And Presentation Improvements:**

- The notations for the length of sentence S and target word T are not being used anywhere in the equations. However, they are still mentioned in line 191-192
- There is an excessive usage of acronyms in the paper which makes it harder to follow

---

> ### Author Rebuttal · Authors · 2023-08-27
>
> We greatly appreciate your diligent review and feedback. The following is our responses to the questions you raised.
>
> A.   \
> Q: (i): how were the baseline models trained and was there any hyperparameter search while training them?
>
> A: For the baseline models presented in Table 3, **we did not train them and directly referenced the reported best results from the papers in which these models were introduced,** as listed in Section 4.2.
> Only the Text-Only Tk-Instruct model was trained by us. We trained it to investigate the effectiveness of our approach in leveraging the visual modality to enhance performance. To ensure controlled variables, we adopted the same hyperparameters as outlined in Section 4.1.
>
> Q: (ii): How were the hyperparameters for the proposed approach (descibed in Section 4.1) set?
>
> A: We experimented with a wide range of hyperparameter combinations, and selected the configuration that exhibited the best performance on the validation set.
>
> Q: (iii): What is the 'ada' model that the authors used for GPT-3 (there are several variants)? Similarly, which ChatGPT model did the authors use (gpt-3.5; gpt-4; which dates? as the models were updated frequently?)
>
> A: GPT-3 has based versions like 'ada', 'babbage', 'curie' and 'davinci'. OpenAI recommends using the 'ada' model for classification tasks in their API reference documentation.\
> We used GPT-3.5 in our paper since the access to the GPT-4 API was restricted by OpenAI at the time of paper submission. To compare with the latest ChatGPT version, **we conducted supplementary experiments using GPT-4 from August 23rd to August 26th. And we will use the following results to update some of the results in Table 7**.
> | Method     | Twitter-2015(Acc)     | Twitter-2015(Macro-F1) | Twitter-2017(Acc)   | Twitter-2017(Macro-F1) |
> | -------- | -------- | -------- | -------- | -------- |
> | GPT-4(zero-shot) | 64.61 | 59.35 | 59.24 | 57.60 |
> | GPT-4(few-shot) | 65.19 | 59.88 | 59.97 | 58.73 |
> | VEMP(GPT-4)(zero-shot) | 65.00 | 59.95 | 59.56 | 58.29  |
> | VEMP(GPT-4)(few-shot) | 65.77 | 61.19 | 60.62 | 59.48  |
>
>
> Q: (iv): How did the authors choose the few-shot examples provided to ChatGPT? Were there any evaluations done to quantify the sensitivity of results to the few-shot examples being used in-context?
>
> A: The **selection criteria** are as follows: (1) Positive, negative, and neutral sentiment samples should all be included. (2) As our classification focused on fine-grained targets, we aimed to have at least two examples using the same input but with different targets and sentiment labels. In Appendix Table 9, Table 11, and Table 13, the second and third examples share the same input but have distinct targets (Rod Petricevic and Bridgecorp) and sentiment labels (negative and neutral). (3) The sentiment labels for each example were uncontroversial. \
> **We tested numerous combinations of different few-shot examples, and the three examples selected in this paper yielded relatively better results** (the result differences were not significant for different few-shot examples).   \
> **The reason to select only 3-shot examples**: Due to the maximum input tokens constraint of Tk-Instruct and the long length of our examples, we were limited to using three examples. And to ensure a fair comparison between our approach and ChatGPT, ChatGPT shared the same 3-shot examples.
>
> Q: B. I am also unsure about the validity of the approach that the authors adopt to extract text from images. It is possible that in some cases the text in the image is split between two different blocks (for instance, in Figure 1, one half of the text is in the top of the image and the other half is in the bottom). Authors mention that they select the bounding box with the maximum area? Does this mean that the authors only consider only half of the text within the image? This would provide incomplete context to the later stages of the approach.
>
> A: As we have explained in lines 223-227: "**Text Symbols Embedded in the Image (TSEI)** from different bounding boxes are usually semantically incoherent if they are spliced together. Besides, TSEI from small bounding boxes are often too brief to provide valuable information, and instead add noises". For instance, in Figure 1(b), only the text "Most points without..." at the top contains useful information, while the text on the player's clothes introduced noises.    \
> We acknowledge that our approach might result in the loss of some useful text within the images. **However, considering the harm introduced by noisy text when incorporating all text from images, we consider our trade-off to be worthwhile. The following supplementary experimental results validate that.**
>
> | Method     | Twitter-2015(Acc)     | Twitter-2015(Macro-F1) | Twitter-2017(Acc)   | Twitter-2017(Macro-F1) |
> | -------- | -------- | -------- | -------- | -------- |
> | No TSEI | 77.82 | 73.70 | 71.56 | 70.77 |
> | All TSEI | 77.34 | 73.43 | 71.23 | 70.55 |
> | Our approach | 78.88 | 75.09 | 73.01 | 72.42  |
>
> **As shown in the above table, due to the negative impact of noises introduced by small bounding boxes, extracting all TSEI can even result in lower performance than not extracting any TSEI. In contrast, our approach (Extraction with Maximum Area Strategy) performs the best.**
>
> Q: C. The authors find that Tk-instruct perform better than ChatGPT (not sure which specific model, as it is not mentioned) but that contary to what has been observed in some recent work (Labrak et al., 2023). This makes me wonder if the authors adopted approaches like Chain of Thought prompting (Wet el al., 2022) to accurately assess the capability of ChatGPT as a baseline. The prompts provided in the appendix do not contain such strategies.
>
> A: Labrak et al.'s work used **Tk-Instruct in zero-shot or few-shot scenarios** for comparison with ChatGPT, whereas we **instruction-tuned Tk-Instruct on the training set**. Hence, our experimental results did not conflict with theirs.    \
> We did not employ the Chain of Thought prompting for ChatGPT. Instead, **we followed the prompt strategy outlined in the paper "Sentiment Analysis in the Era of Large Language Models: A Reality Check" (Zhang et al., 2023, as cited in line 514),** since it was a study of sentiment analysis and closer to our research direction.
>
> Q: Justification For Ethical Concerns:
> The ethics statement of the paper seemed lacking and could have discussed issues around how their approach is influenced by the biased captioning of image and detection of adjective-noun pairs in images. Additionally, the authors do not discuss the data availability and related ethics. They use two existing Twitter datasets and it is unclear if the datasets contain just the Tweet IDs (if that is the case, the data may no longer be available for reproducibility studies or follow-up work).
>
> A: The two Twitter datasets were widely adopted in this field. Prior researchers have provided ethical statements in their papers, ensuring the ethical safety of the datasets. And we employed consistent code for all images to extract image captions and adjective-noun pairs, ensuring objectivity and minimizing bias.    \
> The two publicly available Twitter datasets **contain complete tweet texts and images**. Additionally, the papers cited in Section 4.2 provided the access to these datasets, **and we will also offer that in our released code at a later stage**.
>
> Q: Typos Grammar Style And Presentation Improvements:
> The notations for the length of sentence S and target word T are not being used anywhere in the equations. However, they are still mentioned in line 191-192.
>
> A: We will remove them if you believe it's necessary.
>
> Q: There is an excessive usage of acronyms in the paper which makes it harder to follow.
>
> A: We primarily used four acronyms: Target-oriented Multimodal Sentiment Classification (TMSC),  Visual Elements Mining as Prompts (VEMP), Text Symbols Embedded in the Image (TSEI), and Target-aware Adjective-Noun Pairs (TANPs). The usage of acronyms aims for concise expression as they appeared many times in the paper. If you have any suggestions for improvement, we are open to making revisions.

---

### Official Review · Reviewer_fbv8 · 2023-08-05

**Soundness:** 3

**Excitement:**

3: Ambivalent: It has merits (e.g., it reports state-of-the-art results, the idea is nice), but there are key weaknesses (e.g., it describes incremental work), and it can significantly benefit from another round of revision. However, I won't object to accepting it if my co-reviewers champion it.

**Paper Topic And Main Contributions:**

This paper a Visual Elements Mining as Prompts (VEMP) method, which first maps the useful visual information into texts and leverage the strong instruction tuned language model to finish the task. This paper introduces a novel strategy to utilize the language models.

**Reasons To Accept:**

1. The proposed method is intuitive and easy to understand.
2. The paper is well written.
3. The proposed approach is novel and the results demonstrate its effectiveness.
4. They combine their model with ChatGPT and obtain better results.

**Reasons To Reject:**

They adopt T5 as the text encoder which could result in an unfair comparison.

Update: The authors did not address my concern.  The proposed model utilizes the T5-base(220M) as the encoder which is stronger than BERT-base(110M). However, the baseline models such as FITE adopt BERT-base as the backbone network. It's unfair. So, I will change my score.

**Reproducibility:**

4: Could mostly reproduce the results, but there may be some variation because of sample variance or minor variations in their interpretation of the protocol or method.

**Reviewer Confidence:**

3: Pretty sure, but there's a chance I missed something. Although I have a good feel for this area in general, I did not carefully check the paper's details, e.g., the math, experimental design, or novelty.

---

> ### Author Rebuttal · Authors · 2023-10-19
>
> We apologize for the omission. We did not provide a rebuttal because there was no content included in the "Questions For The Authors" section. We regret that you have lowered the soundness score and excitement score from 4. We hope that the following explanations can alleviate your concern.
> (1) In fact, your concern serves as a strength of our work. As new NLP models continue to emerge, the Target-oriented Multimodal Sentiment Classification (TMSC) field urgently requires innovative approaches that incorporate these new NLP models to drive its development, rather than remaining limited to traditional fine-tuning based on BERT. One of the contributions of our work is to demonstrate the obsolescence of BERT-base within this field and to propose Visual Elements Mining as Prompts (VEMP), that leverage new NLP models to address challenging problems in the field.
> (2) Many baseline models in Table 3 also used stronger encoder. VLP-MABSA used a BART encoder (140M), and it even utilized an additional dataset, MVSA-Multi. Meanwhile, ITM used RoBERTa-base (125M). Therefore, criticizing the inconsistency in encoder choices as a drawback of the paper may not be reasonable. If we aim for absolute fairness, it would imply that all future work should be conducted using BERT-base.
> We would greatly appreciate it if you could restore the scores to 4.

---

### Meta-Review · Area_Chair_f3BP · 2023-09-26

**Recommendation:** 3

**Metareview:**

Most reviewers noted the paper was well written, intuitive and novel/interesting. All of the reviewers had concerns about various experimental design choices not being justified, and one reviewer cited concerns about unfairly comparing the results, but rated the overall soundness of the paper as "good". Nonetheless, after the rebuttal, all of the reviewers found the paper to be adequately sound ("Good").

---

### Decision · Program_Chairs · 2023-10-07

**Decision:**

Accept-Findings

**Comment:**

Most reviewers noted the paper was well written, intuitive and novel/interesting. All of the reviewers had concerns about various experimental design choices not being justified, and one reviewer cited concerns about unfairly comparing the results, but rated the overall soundness of the paper as "good". Nonetheless, after the rebuttal, all of the reviewers found the paper to be adequately sound ("Good").